# HSF3 and Hsp70 Expression during Post-Hatch Cold Stress in Broiler Chickens Subjected to Embryonic Thermal Manipulation

**DOI:** 10.3390/vetsci7020049

**Published:** 2020-04-22

**Authors:** Amneh H. Tarkhan, Khaled M. M. Saleh, Mohammad Borhan Al-Zghoul

**Affiliations:** 1Department of Applied Biological Sciences, Faculty of Science and Arts, Jordan University of Science and Technology, P.O. Box 3030, Irbid 22110, Jordan; amneht92@gmail.com (A.H.T.); khaledmousa93@gmail.com (K.M.M.S.); 2Department of Basic Medical Veterinary Sciences, Faculty of Veterinary Medicine, Jordan University of Science and Technology, P.O. Box 3030, Irbid 22110, Jordan

**Keywords:** broiler, thermal manipulation, *HSF3*, *Hsp70*

## Abstract

Decades of selective breeding for commercial purposes have rendered the broiler chicken (*Gallus gallus domesticus*) highly susceptible to heat and cold stress. A multitude of studies have documented the effects of thermal manipulation (TM) on broiler thermotolerance during periods of post-hatch heat stress, but very few have focused on the effect of TM on a broiler’s ability to withstand cold stress. Therefore, the primary objective of the current study is to determine the effects of TM on the acquisition of thermotolerance in broilers via their expression of the stress-associated 70 kilodalton heat shock protein (*Hsp70*) gene and heat shock factor 3 (*HSF3*) gene. Briefly, Hubbard broiler embryos were subject to TM by increasing the incubation temperature to 39 °C and 65% relative humidity (RH) for 18 h daily, from embryonic days (ED) 10 to 18. Broilers were then exposed to cold stress by decreasing the room temperature to 16 °C during post-hatch days 32 to 37. After thermal challenge, broilers were euthanized and hepatic and splenic tissues were collected. Our results showed that TM decreased the hatchability rate and body temperature but improved the body weight gain. TM generally decreased the hepatic expression but did not change the splenic expression of *HSF3* during cold stress. In contrast, both hepatic and splenic *Hsp70* expression decreased during cold stress. The results of the present study may suggest that TM significantly affects a broiler’s genetic response to cold stress.

## 1. Introduction

A broiler (*Gallus gallus domesticus*) is a meat-type strain of chicken that has been bred and reared solely for the purpose of meat production [1]. Broilers constitute an integral part of the modern human diet, and the rate of chicken consumption is increasing at a faster rate compared to any other meat type [2]. Historically, this was not always the case, as chicken only became a dietary staple after the mid-twentieth century, when the USDA launched the “Chicken of Tomorrow” contests to shift the focus of the poultry industry away from egg and towards meat production [3]. Since then, the poultry industry has exerted many artificial selection pressures on broiler chickens in order to breed for commercially desirable traits, mainly faster growth rates, increased breast meat yield, and improved feed-conversion ratios [4,5]. However, the industry’s narrow focus on body weight has inadvertently resulted in several adverse effects, including immune and metabolic dysfunction as well as increased metabolic costs [6].

Broiler chickens are classified as homoeothermic animals, meaning that they can only maintain their body temperature efficiently when ambient temperatures are within certain limits [7,8]. After decades of selective breeding, commercial broilers produce more body heat yet have an impaired ability to dissipate it, making them highly susceptible to thermal stress [9,10]. Thermal stress can arise at many stages in the typical broiler’s life, and climate change is endangering the welfare and stress levels of broilers around the world [11]. Upon exposure to such stress, broilers have been reported to exhibit a changed expression profile of certain stress-associated genes, including the *HSF3* and *Hsp70* genes [12,13]. The *Hsp70* and *HSF3* genes have been extensively investigated in the context of thermal stress in poultry, as both genes are considered to be a chief defense against heat challenge across many organisms [14,15,16,17].

The 70 kilodalton heat shock proteins (*Hsp70*) are a family of well-conserved molecular chaperones that are constitutively expressed as a part of their housekeeping functions [18]. However, exposure to stress can elevate *Hsp70* expression in response to the increased number of denatured proteins that require refolding [19]. Hsp70 then activates the nuclear factor kappa-B (NF-κB) transcription factor by binding to TLR2 and TLR4, resulting in a signaling cascade with a number of effects [20]. In fact, NF-κB is a focal regulator of inflammatory responses, and its activation induces the expression of the interleukin 6 (IL-6) cytokine [21,22]. In poultry, IL-6 is considered to be a heat-shock gene that is directly activated by heat shock factor 3 (HSF3), the latter of which is also responsible for the induction of Hsp70 in avian cells [23,24].

Thermal manipulation (TM), which involves altering incubation temperature at certain intervals during embryonic development, has been extensively reported to alleviate the effects of heat stress in broilers and enhance muscular growth [25,26,27]. Moreover, TM broilers were found to exhibit a modulated genome-wide gene expression profile upon exposure to heat stress [28]. Since most studies have focused on heat stress, there is a dearth of information regarding the effect of TM during post-hatch cold stress. In addition, two of our previous studies have investigated the effects of the same TM conditions (39 °C and 65% RH for 18 h daily from ED 10 to 18) and subsequent thermal stress on the expression of anti-oxidant and pro-inflammatory genes in the jejunal mucosa, liver and spleen, and the present study will add to those findings [29,30]. Consequently, the aim of the present study is to investigate the effect of TM on post-hatch cold stress on hepatic and splenic *Hsp70* and *HSF3* expression in commercial broilers.

## 2. Materials and Methods

The Animal Care and Use Committee at Jordan University of Science and Technology approved all experimental procedures (approval # 16/3/3/418).

### 2.1. Incubation

Fertile Hubbard eggs (n = 600) were procured from certified breeders based in Irbid, Jordan. After careful examination, eggs with no breakage were then placed in commercial Type-I HS-SF incubators (Barcelona, Spain). Eggs were randomly divided into a control group and a thermal manipulation (TM) group. The control group was incubated at 37.8 °C and 56% relative humidity (RH) during all days of embryonic development (ED), but the TM group was incubated at 39 °C and 65% RH for 18 h/day from ED 10 to 18. For the rest of the incubation period (until ED 21), TM eggs were incubated under standard conditions (37.8 °C and 56% RH) Non-viable eggs were removed on ED 7 after examination by candling.

### 2.2. Hatchery Management

On hatch day, chicks were counted every hour, and, after feather-drying, one-day-old chicks were transported to JUST’s Animal House. From post-hatch days 1 to 37, chicks were randomly distributed into labeled cage pens, and chicks from each group were kept separate. During the first post-hatch week, room temperature was kept constant at 33 ± 1 °C and was gradually lowered to 24 °C by the end of the third week. From post-hatch days 24 to 37, the temperature was maintained at 21 °C. On post-hatch days 1, 3, 5, 7, 9, 11, 13, 15, 19, 22, 25, 28, 30, 33, 35 and 37, the cloacal temperatures (BT) and body weights (BW) of the two incubation groups were recorded for randomly selected chicks (of both genders) (n = 32). Water and feed were supplied ad libitum during the entirety of the experiment. Vaccination against Newcastle disease was carried out on post-hatch days 8 and 20 and vaccination against infectious bursal disease on post-hatch day 15. Figure 1 illustrates the experimental design followed in the present study, which is a modified version of a previously published protocol [29].

### 2.3. Cold Stress

On post-hatch day 32, 120 male chicks from each of the control and TM groups were randomly selected and sub-divided into four sub-groups: controls under normal conditions (CN), controls under cold stress (CC), TM chicks under normal conditions (TN) and TM chicks under cold stress (TC). In order to induce cold stress, the room temperature was lowered to 16 °C from post-hatch day 32 to 37. On post-hatch day 37, a number of chicks (n = 10) were randomly selected from each sub-group to record BT, after which they were humanely euthanized to collect hepatic and splenic samples. All samples were snap-frozen using liquid nitrogen and stored at −80 °C for total RNA extraction. On post-hatch days 32 and 37 (day 0 and day 5 of cold stress, respectively), BW was recorded for each group (n = 10), and the BW gain was calculated for each group as the following: *BW gain* = (BW on day 5 of cold stress) – (BW on day 0 of cold stress).

### 2.4. RNA Isolation and cDNA Synthesis

The Direct-Zol™ RNA MiniPrep (Zymo Research, Irvine, CA, USA) with TRI Reagent^®^ (Zymo Research, Irvine, CA, USA) were employed to extract total RNA from the hepatic and splenic samples. The concentration and purity of isolated RNA were determined on the Biotek PowerWave XS2 Spectrophotometer (BioTek Instruments, Inc., Winooski, VT, USA). From each sample, 2 μg of total RNA was used for cDNA synthesis, the latter of which was performed via the Superscript III cDNA Synthesis Kit (Invitrogen, Thermo Fisher Scientific, Wilmington, NC, USA).

### 2.5. Primer Design

Table 1 lists the primer sequences that were used in the real-time qRT-qPCR analysis. Primer design was carried out on the PrimerQuest tool (https://eu.idtdna.com/pages), and cDNA sequences were obtained from NCBI’s Nucleotide database (https://www.ncbi.nlm.nih.gov/nucleotide/).

### 2.6. Real-Time qRT-qPCR

Relative mRNA quantitation analysis was performed using the QuantiFast SYBR^®^ Green PCR Kit ( QIAGEN Inc., Redwood City, CA, USA) on a Rotor-Gene Q MDx 5 plex instrument (QIAGEN Inc., Redwood City, CA, USA). Briefly, 10 µL of master mix, 1.2 µL of forward primer, 1.2 µL of reverse primer, 1 µL of sample cDNA and 6.6 µl of nuclease-free water were combined to create 20 µL of reaction mix. PCR cycles were set at the following parameters: 5 min at 95 °C; 40 cycles of 95 °C for 10 s and 55 °C for 30 s; and 10 s at 72 °C with a final melting phase at 95 °C for 20 s. During the extension step, fluorescent emission was detected. The 28S ribosomal RNA, an internal control, was the reference to which fold changes in gene expression were normalized. Triplicates from each cDNA library were analyzed and the single target amplification specificity was approved by the melting curve. Relative quantitation was determined automatically.

### 2.7. Statistical Analysis

IBM SPSS Statistics v.23 (IBM, USA) was utilized for all the statistical analyses performed in the current study. Hatchability rates were analyzed using the chi-squared (χ^2^) test. BT, BW and mRNA levels of the *HSF3* and *Hsp70* genes were expressed as means ± SD. For each experimental time interval (post-hatch days 1, 3, 5, 7, 9, 11, 13, 15, 19, 22, 25, 28, 30, 33, 35 and 37), comparison between different parameters of the TM and control groups was carried out by means of an independent *t*-test. To compare between different treatment groups of Cold stress experiment, one-way ANOVA followed by Bonferroni tests were employed. Parametric differences were statistically significant if the *p*-value < 0.05.

## 3. Results

### 3.1. Effect of Embryonic Thermal Manipulation (TM) on Physiological Parameters

TM led to a significantly lower hatchability rate in comparison with controls (Table 2). Furthermore, TM led to significantly lower body temperatures (BT) in broiler chicks on all experimental time intervals except for post-hatch day 5, 15 and 37 (Figure 2). Additionally, TM significantly increased the body weight (BW) of broiler chicks only on post-hatch days 35 and 37 (Figure 3).

### 3.2. Effect of Post-Hatch Cold Stress (CS) on the Physiological Parameters

Figure 4A,B shows the effects of post-hatch CS on the BW and BT of TM broiler chicks. TM did not significantly change the BT of broiler chicks kept under normal conditions (Figure 4A), but exposure to CS caused TM chicks to have significantly lower BT in comparison with controls. Furthermore, no significant changes were detected in BW between all groups on days 0 and 5 of CS (Figure 4B). However, CS was found to significantly decrease BW gain in both the TM and control groups.

### 3.3. Effect of Post-Hatch Cold Stress (CS) on the mRNA Levels of HSF3

Figure 5A,B represents the effects of post-hatch cold stress on the mRNA levels of *HSF3* in the liver and spleen of TM broiler chicks.

#### 3.3.1. Hepatic Expression

Compared to the CS controls, *HSF3* mRNA levels were significantly lower in the CS TM group, but no significant difference was observed between the control and TM groups under normal conditions.

#### 3.3.2. Splenic Expression

Compared to all other groups, the mRNA levels of *HSF3* were significantly higher in the TM group under normal conditions, however, no significant difference was detected between the other three groups.

### 3.4. Effect of Post-Hatch Cold Stress (CS) on the mRNA Levels of Hsp70

Figure 6A,B represents the effects of post-hatch cold stress on the mRNA levels of *Hsp70* in the liver and spleen of TM broiler chicks.

#### 3.4.1. Hepatic Expression

Compared to all other groups, *Hsp70* mRNA levels were significantly increased in CS controls, but no significant difference was detected between the other three groups.

#### 3.4.2. Splenic Expression

The mRNA levels of *Hsp70* were significantly increased in CS controls compared to non-CS controls and CS TM chicks. However, no significant difference was observed between the non-CS controls and non-CS TM chicks, nor between the non-CS TM chicks and CS TM chicks.

## 4. Discussion

The modern broiler is especially vulnerable to the effects of thermal stress, the latter of which can cause a range of commercial losses. Moreover, thermal stress in avian cells can alter the expression of several genes, including those associated with the stress response. The main objective of the present study was to investigate the effects of thermal manipulation (TM) on heat shock gene expression, namely *HSF3* and *Hsp70*, during post-hatch cold stress (CS).

In the current study, TM was achieved by raising the incubation conditions to 39 °C and 65% RH for 18 h/day from ED 10 to 18. Previous studies used similar TM treatment parameters, including incubation at 39 °C for 18 h/day from ED 7–11, at 38.5 °C from ED 16 to 18 and at 39 °C for 18 h/day from ED 12 to 18, and these parameters were shown to significantly enhance BW as well as upregulate expression of muscle marker genes [27,31,32,33]. The duration of TM was reported to affect thermotolerance acquisition in broilers, and different TM treatments led to differences in physiological parameters as well as varied mRNA expression profiles of heat shock genes [34,35,36,37].

Broiler hatchability can be affected by a number of different factors, including breed, age, incubation conditions and season [38]. Compared to controls, TM chicks were found to have a reduced hatchability rate in the current study. In other avian species like ostriches and Japanese quail, TM resulted in a similarly decreased hatchability rate [39,40]. However, mixed findings were reported for the effect of TM on broiler hatchability, as some studies found that TM positively or negatively affected the hatchability rate [31,41,42,43], while others reported no effect [36,43,44,45,46]. Such variation in hatchability rates between different TM studies could be due to differences in age, breed and TM treatment conditions.

Changes to the ambient temperature can negatively affect the performance of broilers, causing them to alter their behavior in an attempt to reduce their body temperature (BT) [47]. In the present study, BT was reduced in TM chicks compared to controls during CS. However, TM was not shown to significantly affect the facial surface temperature of Ross 308 broilers before and during chronic HS [48].HS has been reported to affect the body weight (BW) of broilers by decreasing their feed intake and efficiency [49]. In contrast, our findings show that broiler BW was found to be higher in TM chicks compared to controls later in their post-hatch lives (post-hatch days 35 and 37). Corresponding with our results, reports have shown that TM led to a generally higher BW than in controls near marketing age [32]. Moreover, TM has been shown to enhance myoblast proliferation and pectoral muscle weights in broilers during their post-hatch life [31,50].

The *HSF3* gene is unique to avian and reptilian cells, and it is activated during extended periods of heat stress [23,51,52]. HSF3 is essential to the induction and regulation of *Hsp70* expression by directly binding to the c-myb proto-oncogene product, the latter of which is involved in cellular proliferation [53]. In the present study, during CS, hepatic *HSF3* expression was lower in TM chicks, but splenic expression did not differ between TM and control chicks. Previously, TM was found to enhance *HSF3* expression during HS in Cobb chicks, but it did enhance basal *HSF3* expression in TM chicks compared to controls under normal conditions [34]. In other studies, TM was found to increase *HSF3* expression in Cobb and Hubbard chicks during post-hatch HS [13,35].

The *Hsp70* gene is induced during times of cellular stress, and it functions mainly to protect against cell death and the effects of protein aggregation [54]. Within the cell, Hsp70 acts as a chaperone. Our findings show that CS decreased hepatic and splenic *Hsp70* expression, respectively. Previous reports have shown that, during HS, *Hsp70* expression was found to be lower in a commercial broiler breed (Cobb) compared to local ones (Caneluda and Peloco), and this high expression level was associated with resistance to elevated temperatures in the local breeds [12]. Moreover, Cobb and Hubbard chicks subjected to TM were found to express higher cardiac, hepatic, muscular and splenic levels of *Hsp70* during post-hatch heat challenge [13,34,35,42]. The differences between *Hsp70* expression in control and TM chicks may be attributed to modulated DNA methylation of the *Hsp70* promoter [55].

The strengths of the present study lie in the fact that all chicks were of the same age, strain (Hubbard) and sex (male). In addition, any thermal stress that could arise due to non-experimental conditions was mitigated by having the rearing and experimental rooms in close proximity to one another. In contrast, some limitations of the present study include the relatively small sample size and the fact that the effect of TM on embryonic mortality and physiological parameters was not fully explored. However, the sample sizes utilized in the current study are in line with previously published reports [12,29].

## 5. Conclusions

The findings of the present study suggest that TM of broilers during certain intervals of their embryonic development may alter heat shock genes expression and the response to post-hatch cold stress. This suggest that pre-hatch TM may possess a long-lasting impact on the response to different types of stress (not only to heat stress). Thus, TM may cause a significant improvement in broilers production by enhancing their growth and performance under post-hatch severe conditions.

## Figures and Tables

**Figure 1 vetsci-07-00049-f001:**
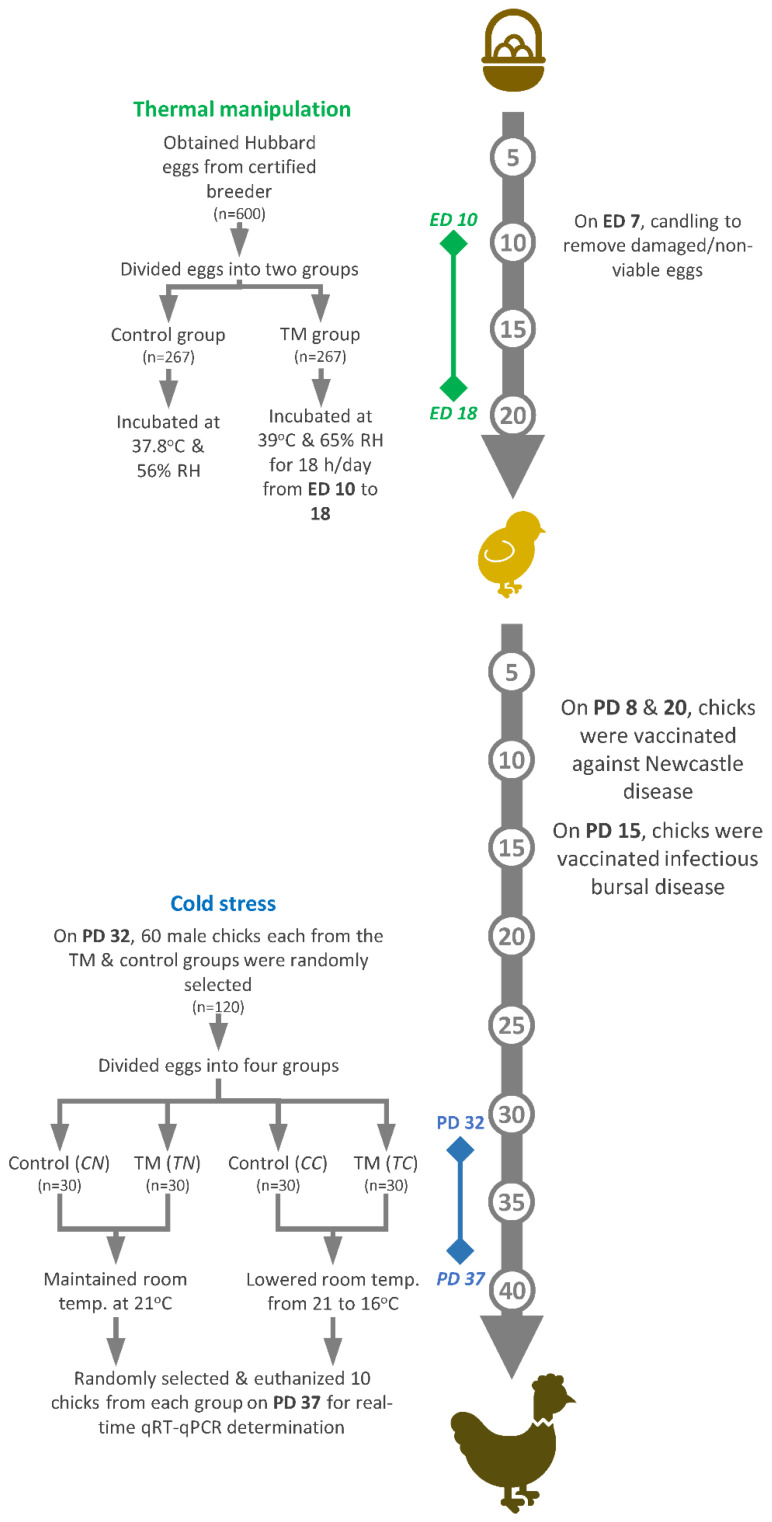
The experimental design constitutes two key components: thermal manipulation and cold stress. ED: embryonic day; PD: post-hatch day.

**Figure 2 vetsci-07-00049-f002:**
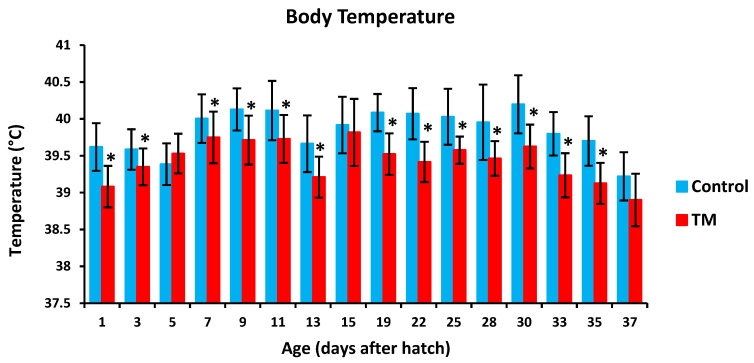
Effect of thermal manipulation (TM) on post-hatch body temperature (BT) of broiler chicks. * within the same day, differences in means ± SD of TM and control chicks are significant (*p* < 0.05).

**Figure 3 vetsci-07-00049-f003:**
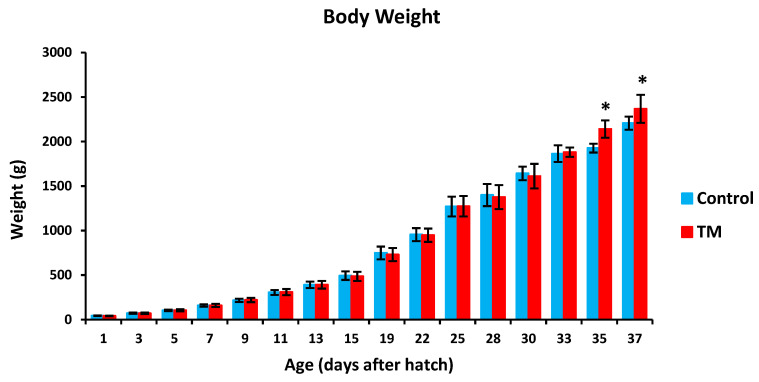
Effect of thermal manipulation (TM) on post-hatch body weight (BW) of broiler chicks. * within the same day, differences in means ± SD of TM and control chicks are significant (*p* < 0.05).

**Figure 4 vetsci-07-00049-f004:**
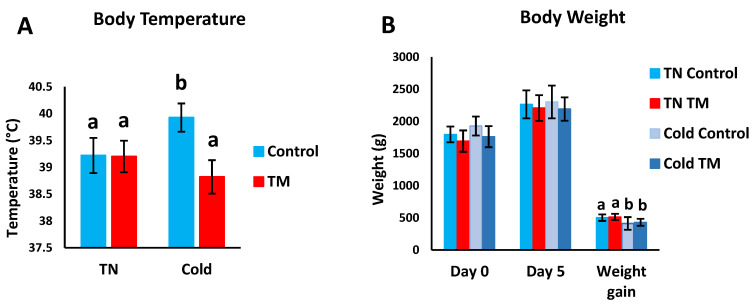
Effect of post-hatch cold stress at 16 °C for 5 days (post-hatch days 32 to 37) on body temperature (BT); body weight (BW) and BW gain. Panel (**A**) represents the BT on day 5 of cold stress (post-hatch day 37). Panel (**B**) represents the BW on day 0 and day 5 of cold stress (post-hatch days 32 and 37, respectively) and the BW gain during the 5 days of cold stress. ^a,b^ means ± SD with non-identical superscripts are significantly different (*p* < 0.05).

**Figure 5 vetsci-07-00049-f005:**
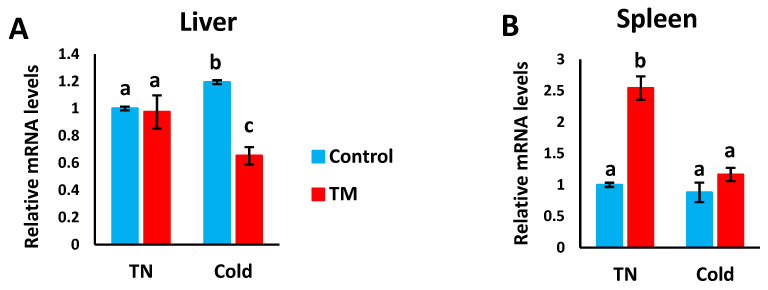
Effect of post-hatch cold stress at 16 °C for 5 days (post-hatch days 32 to 37) on the (**A**) fold changes of *HSF3* mRNA in the liver and (**B**) fold changes of *HSF3* mRNA in the spleen. ^a,b,c^ means ± SD with non-identical superscripts are significantly different (*p* < 0.05).

**Figure 6 vetsci-07-00049-f006:**
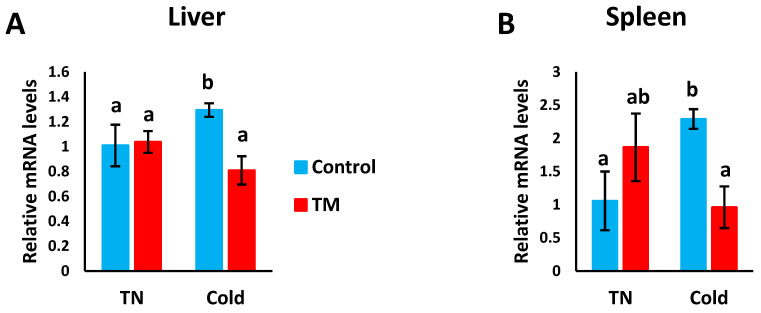
Effect of post-hatch cold stress at 16 °C for 5 days (post-hatch days 32 to 37) on the (**A**) fold changes of *Hsp70* mRNA in the liver and (**B**) fold changes of *Hsp70* mRNA in the spleen. ^a,b^ means ± SD with non-identical superscripts are significantly different (*p* < 0.05).

**Table 1 vetsci-07-00049-t001:** Forward and reverse primer sequences for the qRT-qPCR analysis.

Gene	Sequence (5′ to 3′)
28S rRNA ^1^	F: CCTGAATCCCGAGGTTAACTATT R: GAGGTGCGGCTTATCATCTATC
*HSF3*	F: TTAGAGAGGTTGGAGGGTATGA R: GAATCTGCTCGAGGCGTATAG
*Hsp70*	F: AGAGGAAACTGTGACCCGATGA R: AACGAAGAGGAAGATGGCGA

^1^ Internal control.

**Table 2 vetsci-07-00049-t002:** Effect of thermal manipulation (TM) on broiler hatchability.

Parameter	Control	TM
Total eggs	266	268
Hatched eggs	248	234
Hatchability	93.23% ^a^	87.31% ^b^

^a,b^ within the same row, non-identical superscripts indicate significant differences.

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
