# Peer review of "HSF3 and Hsp70 Expression during Post-Hatch Cold Stress in Broiler Chickens Subjected to Embryonic Thermal Manipulation"

_vetsci, 2020, doi:10.3390/vetsci7020049_

Round 1
Reviewer 1 Report
The study is of great significance considering that broilers are reared in many regions with varying climatic conditions, and sometimes poor environmental control during incubation and grow-out period. This study, albeit ethically questionable with regards to how high or low temperatures can be taken, provides room for improvement when discrepancies occur at incubation and post-hatch.
However, the experimental designed needs to be revisited and precisely described. There are steps in Figure 1 that are not agreeable with text in the Materials and Methods, creating confusion for the reader.
It is also not clear why the heat stress and cold stress treatments were applied at different ages or periods. Obviously, this influences broiler response physiologically.
Please see further comments in the document.

Author Response
The study is of great significance considering that broilers are reared in many regions with varying climatic conditions, and sometimes poor environmental control during incubation and grow-out period. This study, albeit ethically questionable with regards to how high or low temperatures can be taken, provides room for improvement when discrepancies occur at incubation and post-hatch.
However, the experimental designed needs to be revisited and precisely described. There are steps in Figure 1 that are not agreeable with text in the Materials and Methods, creating confusion for the reader.
It is also not clear why the heat stress and cold stress treatments were applied at different ages or periods. Obviously, this influences broiler response physiologically.
Answer: to remove the confusion in the experimental design, the chronic heat stress experiment was removed as we are planning to submit it as a separate manuscript.
- Extensive editing of English language and style required
Answer: the manuscript has been revised with track changes.
- Line 21: Considering that it's during incubation, why not refer to as "embryos"? Please rephrase sentence.
Answer: the sentence had been rephrased to show that the treatment was on embryos.
- Lines 22 to 24: Was this within the 21-day incubation period at d 18 to 21, or post-hatch? Please be concise.
Answer: the sentence is rephrased; the treatment was during post-hatch days 32 to 37.
- Lines 79 to 81: Were eggs incubated for 18 or 21 days? Please be specific that embryonic development was over 18 days of incubation. however, if incubation lasted for 21 days, it must be state clearly, and whether temperature and humidity were changed to those in the Control from day 18 to 21.
Answer: the sentence is rephrased as the following:
The control group was incubated at 37.8 °C and 56% relative humidity (RH) during all days of embryonic development (ED), but the TM group was incubated at 39 °C and 65% RH for 18 h/day from ED 10 to 18; and the rest of incubation period (until ED 21) was under the standard conditions (as the conditions of the control group).
- Lines 83 to 84: Were both sexes used in the experiment?
- Line 94: When was sexing done on the chicks as this was not mentioned post-hatch or during placement when chicks were randomly allocated to the two groups?
Answer: both sexes were used for body weight and temperature monitoring during normal post-hatch conditions; but only male chicks were selected and used in the cold stress experiment.
- Line 100: At day 27 above, the temperature was at 21 degrees C. Is this correct? Unless the rest of the group was not subjected to the heat stress and cold stress treatments, respectively.
- Line 103: To read: Chicks
- Lines 104 to 105: Transported or transfered (or relocated)?
- Line 105: Were d 26 to 27 used as adaptation period for the chicks? What were the environmental conditions during these two days for this group?
- Line 112: Why were different periods or days used for the heat stress and cold stress treatments? Age has changed, so cold stress effects will likely be different compared to a broiler at 26 d of age, because the level of fat deposition is different. Does this not influence the hypothesis and/or objective for the study.
Answer: This section of the manuscript, the part of heat stress, is deleted.
- Lines 156 to 157: Were BW at hatch or 1-d-old not recorded to see differences in initial BW and how they changed and compared at d 7.
Answer: BW has been recorded on day of hatch (1-d) and other post hatch days (3, 5, 7, 9, 11, 13, 15, 19, 22, 25, 28, 30, 33, 35, and 37), but only days 35 and 37 showed a significant difference in the BW between TM and control groups.
- Line 169: What could be the reason for the high BT in the control broilers compared to those in the TM group, which had higher incubation temperature and humidity levels? Does this imply that the control birds are likely to suffer from heat stress under high environmental temps?
- Line 176: There seems to be a huge difference between D0 and D1, then D1 and D3 for the same birds in both the control & TM groups.
Answer: the part of heat stress is deleted from the manuscript.
- Line 251: Broiler chicks are more sensitive to temperature and humidity between 1 and 21 days of age, especially at lower values. Then the reverse is true as they get older and accumulate more muscle and fat. How was this background factored in the experiment?
Answer: during all experiment intervals, the chicken were reared under controlled temperature and humidity rooms to ensure all background factors are similar between different treatments and did not influence the results.
- Lines 264 to 266: How do these results compare to breed standards under normal environmental conditions during incubation and at post-hatch?
Answer: based on the HUBBARD management guide, our results of the hatchability rate, body weight gain are comparable.
- Line 283: What could be the reason for these flactuations?
Answer: the results of HS were removed as we are intended to submit as a separate manuscript.
- Lines 299 to 300: However, the heat stress and cold stress treatments were applied at different stages of growth. What limitation does this create?
Answer: the part of heat stress is deleted
- Lines 309 to 311: What are the implications of this conclusion on production parameters and grow-out periods?
Answer: The findings of the present study suggest that TM of broilers during certain intervals of their embryonic development may alter heat shock genes expression and the response to post-hatch cold stress. This suggest that pre-hatch TM may possess a long-lasting impact on the response to different types of stress (not only to heat stress). Thus, TM might cause a significant improvement in broilers production by enhancing their growth and performance under post-hatch severe conditions.
Reviewer 2 Report
This manuscript is a continuation of the author's past series research in avian embryonic thermal manipulation (TM). The main breakthrough is to further observe the response of TM broilers to cold stress during their growth. It does provide an issue that has not been noticed in the past. The results of this study have scientific merit and application value. However, there are some questions as following:
- The experimental design has incomprehensible flaws. A simple two by three factorial design could have been used. The time interval for clod stress (d32-37) and heat stress (d28-35) was different which leads to the inconsistency of the data display.
- The way of results presented for "effect of post-hatch heat stress" (3-2~3-4, line 165-205) are not the same as the results presented for "effect of post-hatche cold stree" (3-5~3-7, line206-245. It can be understood that the way cold stress data were shown may be due to it's overlap period with the HS (d32-d35) only falls in to these days. If this is the reason, then the title of the 3.5 and 3.6 shoud be changed to include the HS effect but not only the cold stress effect.
- Figure 6(A) did not provide the day of BW taken.
Author Response
This manuscript is a continuation of the author's past series research in avian embryonic thermal manipulation (TM). The main breakthrough is to further observe the response of TM broilers to cold stress during their growth. It does provide an issue that has not been noticed in the past. The results of this study have scientific merit and application value. However, there are some questions as following:
- The experimental design has incomprehensible flaws. A simple two by three factorial design could have been used. The time interval for clod stress (d32-37) and heat stress (d28-35) was different which leads to the inconsistency of the data display.
- The way of results presented for "effect of post-hatch heat stress" (3-2~3-4, line 165-205) are not the same as the results presented for "effect of post-hatche cold stree" (3-5~3-7, line206-245. It can be understood that the way cold stress data were shown may be due to it's overlap period with the HS (d32-d35) only falls in to these days. If this is the reason, then the title of the 3.5 and 3.6 shoud be changed to include the HS effect but not only the cold stress effect.
Answer: The part of heat stress is deleted
Figure 6(A) did not provide the day of BW taken.
Answer: the caption is changed as the following: Figure 3. Effect of post-hatch cold stress at 16 °C for 5 days (post-hatch days 32 to 37) on the body temperature (BT); body weight (BW) and BW gain. Part (a) represent the body temperature (BT) of different cold stress groups on the day 5 of cold stress (post-hatch day 37). Part (b) represents the body weight (BW) of different cold stress groups on the day 0 and day 5 of cold stress (post-hatch days 32 and 37, respectively); and the BW gain. a,b means ± SD with non-identical superscripts are significantly different (p < 0.05).
This manuscript is a resubmission of an earlier submission. The following is a list of the peer review reports and author responses from that submission.
Round 1
Reviewer 1 Report
The main aim of this study was to investigate the effects of thermal stress on the hepatic and splenic mRNA levels of Hsp70and HSF3 genes. They showed that thermally manipulated broilers exhibited an enhanced expression profile of the Hsp70 and HSF3 genes, a result that might improve the acquisition of thermo tolerance. According to the subsequent issues, we would suggest the small size of sample amount is not enough to support their findings.
General comments
In terms of thermal incubation could change the hatching rate, that means maybe some weak embryos died under the impacts of thermal incubation, so that the left embryos are stronger ones, and they ultimately developed to a population with better physiology states and competition activities in lives. I would say that maybe the reason of why differences were observed in body temperature and weight. How do you think about that?
The research only adopted two genes for qPCR determination, why chose these two genes for analysis? I don’t think it is enough for explaining the hepatic molecular changes very well under the temperature treatments.
The sample size for cold treatments is only five for each sub-group? I thought the sample number is too small to support their conclusions. And they did not list the sample size for qPCR determination.
Reviewer 2 Report
I was just wondering why the heat stress and cold stress sub-experiments were conducted differently.
The heat stress experiment had chicks from the control and TM groups exposed to HS over a period of 7 days. The cold stress experiment had chicks from control and TM groups further divided into TN and CS groups, and then exposed to cold stress for 5 days. Moreover, the chicks are exposed to HS from PD 28-35 whereas the other group of chicks is exposed to CS from PD 32-37.This study design for both these sub-experiments isn't justified/explained in the introduction nor methods section. A reasoning for why these two were conducted differently with respect to duration of PD thermal stress, specific time of thermal stress at PD age, and potentially why certain control groups were/not included, might help the reader better understand the biological significance of the changes in abundance of the genes evaluated.
Overall, an interesting study that is easy to follow the way it is written.